# Sharing Knowledge for Meta-learning with Feature Descriptions

**Tomoharu Iwata**
NTT Communication Science Laboratories
Kyoto, Japan
`tomoharu.iwata.gy@hco.ntt.co.jp`

**Atsutoshi Kumagai**
NTT Computer and Data Science Laboratories
Tokyo, Japan
`atsutoshi.kumagai.ht@hco.ntt.co.jp`

## Abstract

Language is an important tool for humans to share knowledge. We propose a meta-learning method that shares knowledge across supervised learning tasks using feature descriptions written in natural language, which have not been used in the existing meta-learning methods. The proposed method improves the predictive performance on unseen tasks with a limited number of labeled data by meta-learning from various tasks. With the feature descriptions, we can find relationships across tasks even when their feature spaces are different. The feature descriptions are encoded using a language model pretrained with a large corpus, which enables us to incorporate human knowledge stored in the corpus into meta-learning. In our experiments, we demonstrate that the proposed method achieves better predictive performance than the existing meta-learning methods using a wide variety of real-world datasets provided by the statistical office of the EU and Japan.

## 1 Introduction

Humans can learn new tasks using descriptions written in natural language, which support us in improving the performance on the new tasks especially when we have little experience since the descriptions help understand how experience on familiar tasks can be used for the new tasks. For example, Table 1 shows a dataset from Eurostat, which is the statistical office of the European Union. Some people can guess the production value at basic price in Denmark on soft wheat and spelt in 2017 given Table 1 using knowledge about countries, cereals, and agriculture. On the other hand, supervised learning methods using data shown in Table 1 cannot guess the value since some features, i.e., 'Denmark' and 'soft wheat and spelt', do not appear in the training data. Meta-learning methods have been proposed for sharing knowledge across supervised learning tasks to improve the performance on unseen tasks with a small amount of training data. However, the existing meta-learning methods do not use feature descriptions written in natural language although many real-world datasets contain feature descriptions as shown in Table 1.

In this paper, we propose a meta-learning method that learns from datasets with feature descriptions to improve the generalization performance on unseen tasks with small labeled data. With feature descriptions, we can share knowledge between different tasks with similar descriptions even when their feature spaces are different. We transform the feature descriptions to sentence embeddings by a sentence encoder, e.g., Bidirectional Encoder Representations from Transformers (BERT) [3], pretrained with a large-scale corpus, by which we can exploit a huge amount of human knowledge accumulated in the corpus for meta-learning. The sentence embeddings reflect the meaning of the descriptions but they are not necessarily suitable for supervised learning tasks. We translate them by a feature encoder network to improve the generalization performance.

We embed instances in a task-specific space by an instance encoder network using their feature values and descriptions translated into sentence embeddings. By adapting a Gaussian process (GP) [17] to

Table 1: Example of a dataset with feature descriptions in Eurostat on economic accounts for agriculture. In this paper, *feature descriptions* denote explanations about features for each instance, e.g., 'Production value at basic price' and 'Wheat and spelt', and *feature type descriptions* denote explanations about types of the features shown at the top row, e.g., 'Agricultural indicator ' and 'List of products'. The 'Label' column comprises labels to be predicted.

| Agricultural indicator | List of products | Unit of measure | Geo | Time | Label |
|---|---|---|---|---|---|
| Production value at basic price | Wheat and spelt | Million EURO | Belgium | 2016 | 189.66 |
| Production value at basic price | Wheat and spelt | Million EURO | Belgium | 2017 | 237.56 |
| Production value at basic price | Barley | Million EURO | Italy | 2016 | 158.82 |
| Subsidies on products | Cereals (including seeds) | Million EURO | France | 2016 | 6.19 |
| Tax on products | Industrial crops | Million EURO | Germany | 2016 | 17.00 |

small labeled data in the space, we obtain a task-specific prediction model. The neural networks for feature encoding, instance encoding, and prediction are shared among all tasks, which enables us to learn and share knowledge that is useful in different tasks. The parameters of the neural networks are trained by minimizing the expected test error of the task-adapted prediction model using meta-training datasets based on a bilevel optimization framework. The meta-training datasets consist of labeled data from a wide variety of tasks with feature descriptions.

The main contributions of this paper are as follows: 1) We newly present a meta-learning problem where feature descriptions are given for each task. 2) We propose a simple yet effective neural network-based meta-learning method that improves the generalization performance with small labeled data using feature descriptions by learning from various datasets. 3) We empirically demonstrate that the proposed method outperforms the existing methods using a wide variety of real-world datasets provided by the statistical office of the EU and Japan.

## 2   Related work

Many meta-learning methods have been proposed [19, 1, 4, 26, 20]. However, they usually assume that only pairs of feature and label values are given for each task, and do not use task information. Several existing meta-learning methods use task information, such as class descriptions and task descriptions. Class descriptions, e.g., image category name in natural language, have been used for few-shot [30] and zero-shot learning [5, 21]. Task descriptions, e.g., the brightness of images in the task, have been used for active meta-learning [9], few-shot learning [13], zero-shot domain adaptation [31], and natural language processing systems [28]. However, they cannot use feature descriptions. Feature descriptions are helpful for understanding relationships between tasks especially when feature spaces are different across tasks. Most existing meta-learning methods assume feature spaces are the same for all tasks although tasks with different feature spaces can contain useful knowledge to be shared. Heterogeneous meta-learning can handle tasks with different feature spaces [7]. However, it cannot use the information about features. Text information has been used in some applications with supervised learning settings, such as cross-lingual text classification [14], recommendation [27], and image classification [6]. However, they are inapplicable to meta-learning. We use text information for sharing knowledge to improve performance on unseen tasks.

## 3   Proposed method

### 3.1   Problem formulation

In the meta-training phase, we are given $D$ labeled meta-training datasets with feature descriptions, $\mathcal{D} = \{\{(\mathbf{x}_{dn}, y_{dn})\}_{n=1}^{N_d}, \mathbf{W}_d\}_{d=1}^{D}$, where $\mathbf{x}_{dn} = (x_{dn1}, \ldots, x_{dnJ_d})$ is the feature vector of the $n$th instance, $y_{dn} \in \mathbb{R}$ is its label, $\mathbf{W}_d = \{\{\mathbf{w}_{djc}\}_{c=1}^{C_{dj}}\}_{j=1}^{J_d}$, is the set of feature descriptions, $N_d$ is the number of instances, $x_{dnj} \in \{1, \ldots, C_{dj}\}$ is the categorical value of the $j$th feature, $C_{dj}$ is the number of categories, $J_d$ is the number of features, and $\mathbf{w}_{djc}$ is the feature description of the $c$th category of the $j$th feature. For example, $\mathbf{w}_{d11} =$ 'Production value at basic price', $\mathbf{w}_{d12} =$ 'Subsidies on products', and $\mathbf{w}_{d21} =$ 'Wheat and spelt' are feature descriptions of the dataset in Table 1. The feature description sets $\mathbf{W}_d$, the number of features $J_d$, and the number of categories $C_{dj}$ are different across tasks. In the meta-test phase, we are given a support set, which consists of a small number of

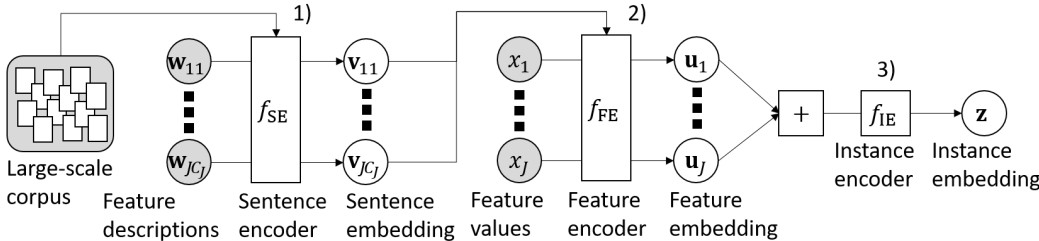

Figure 1: Our encoder. 1) Feature descriptions $\{\{\mathbf{w}_{jc}\}_{c=1}^{C_j}\}_{j=1}^{J}$ are transformed to sentence embedding vectors $\{\{\mathbf{v}_{jc}\}_{c=1}^{C_j}\}_{j=1}^{J}$ using sentence encoder $f_{\mathrm{SE}}$ pretrained with a large-scale corpus. 2) Feature values $\{x_j\}_{j=1}^{J}$ are transformed to feature embedding vectors $\{\mathbf{u}_j\}_{j=1}^{J}$ by feature encoder $f_{\mathrm{FE}}$ using the sentence embedding vectors. 3) Instance embedding vector $\mathbf{z}$ is obtained by encoding the average of the feature embedding vectors with instance encoder $f_{\mathrm{IE}}$. Shaded and unshaded circles represent observed and hidden variables, respectively.

labeled examples $\mathbf{D}^{\mathrm{S}} = \{(\mathbf{x}_n^{\mathrm{S}}, y_n^{\mathrm{S}})\}_{n=1}^{N^{\mathrm{S}}}$ with description $\mathbf{W}$ in an unseen task, which are different from the meta-training datasets. Our aim is to improve the test label prediction performance in test tasks. Although we explain the proposed method for categorical features and numerical labels for simplicity, it can also use numerical features and categorical labels as described in Section 3.4.

## 3.2 Model

With our model, instances with feature descriptions are embedded for each task as described in Section 3.2.1, and their labels are predicted in the task-specific embedding space as described in Section 3.2.2.

### 3.2.1 Instance embedding with feature descriptions

We embed each instance $\mathbf{x} = (x_1, \ldots, x_J)$ using feature descriptions $\mathbf{W} = \{\{\mathbf{w}_{jc}\}_{c=1}^{C_j}\}_{j=1}^{J}$ as shown in Figure 1, where $x_j \in \{1, \ldots, C_j\}$ is the $j$th feature value, $J$ is the number of features, $C_j$ is the number of categories of the $j$th feature, and $\mathbf{w}_{jc}$ is the feature description of the $c$th categorical value at the $j$th feature. Here, we omit the task and instance indices.

First, each feature description $\mathbf{w}_{jc}$ is transformed into sentence embedding $\mathbf{v}_{jc} \in \mathbb{R}^{M_{\mathrm{v}}}$ by a pretrained sentence encoder, e.g., BERT,

$$\mathbf{v}_{jc} = f_{\mathrm{SE}}(\mathbf{w}_{jc}) \tag{1}$$

where $f_{\mathrm{SE}}$ is the sentence encoder. With BERT pretrained using a huge number of text documents, we can incorporate knowledge in the documents.

Second, feature embedding $\mathbf{u}_j \in \mathbb{R}^{M_{\mathrm{u}}}$ of the $j$th feature value is obtained using the sentence embedding and feature value,

$$\mathbf{u}_j = f_{\mathrm{FE}}(\mathbf{v}_{jx_j}), \tag{2}$$

where $f_{\mathrm{FE}}$ is a trainable feature encoder, such as a feed-forward neural network and transformer [25]. By the feature encoder, we translate sentence embeddings, which are calculated based on the pretraining documents written for humans, to feature embeddings that are suitable for supervised learning tasks.

Third, instance embedding $\mathbf{z} \in \mathbb{R}^{M_{\mathrm{z}}}$ is obtained by averaging the feature embeddings,

$$\mathbf{z} = f_{\mathrm{IE}}\left(\frac{1}{J}\sum_{j=1}^{J}\mathbf{u}_j\right) \equiv f(\mathbf{x}; \mathbf{W}), \tag{3}$$

where $f_{\mathrm{IE}}$ is a trainable instance encoder based on neural networks, and $f$ is an encoder to output instance embedding $\mathbf{z}$ given feature vector $\mathbf{x}$ and feature description $\mathbf{W}$. Eq. (3) is a permutation

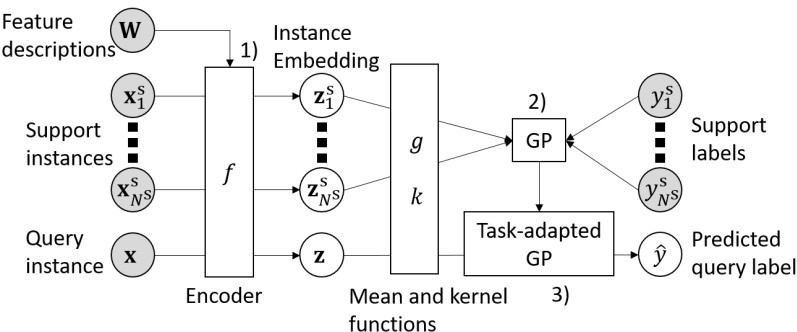

Figure 2: Our task-specific prediction model. 1) Feature vectors of support $\{\mathbf{x}_n^{\mathrm{S}}\}_{n=1}^{N^{\mathrm{S}}}$ and query $\mathbf{x}$ instances are transformed to instance embeddings $\{\mathbf{z}_n\}_{n=1}^{N^{\mathrm{S}}}$ and $\mathbf{z}$ by encoder $f$ in Eq. (3) with feature descriptions $\mathbf{W}$. 2) Using support instance embeddings $\{\mathbf{z}_n\}_{n=1}^{N^{\mathrm{S}}}$ and their labels $\{y_n^{\mathrm{S}}\}_{n=1}^{N^{\mathrm{S}}}$, the GP with mean function $g$ and kernel function $k$ is adapted. 3) Predicted query label $\hat{y}$ is obtained with the task-adapted GP and its instance embedding $\mathbf{z}$. Shaded and unshaded circles represent observed and hidden variables, respectively. Encoder $f$ and mean $g$ and kernel $k$ functions are shared across tasks.

invariant neural network [32] since the summation is invariant even when the elements are permuted, and our model can handle instances with any number of features. Neural networks, $f_{\mathrm{FE}}$ and $f_{\mathrm{IE}}$, are shared across all tasks, by which we can store knowledge meta-learned in various tasks in the neural networks, and use it for unseen tasks. Although size $J$ of input feature vector $\mathbf{x}$ can be different across tasks, size $M_{\mathrm{z}}$ of instance embedding vector $\mathbf{z}$ is the same for all tasks.

### 3.2.2 Task-specific prediction

The proposed model predicts label $y$ of query instance $\mathbf{x}$ using support set $\mathbf{D}^{\mathrm{S}} = \{(\mathbf{x}_n^{\mathrm{S}}, y_n^{\mathrm{S}})\}_{n=1}^{N^{\mathrm{S}}}$ and feature description $\mathbf{W}$ of the task, where $\mathbf{x}_n^{\mathrm{S}}$ is the feature vector of the $n$th support instance, $y_n^{\mathrm{S}}$ is its label, and $N^{\mathrm{S}}$ is the number of support instances. Here, we omit the task index. Figure 2 illustrates our model.

First, instance embeddings are obtained by $f$ in Eq. (3) for the query and support instances, $\mathbf{z} = f(\mathbf{x}, \mathbf{W})$, and $\mathbf{z}_n^{\mathrm{S}} = f(\mathbf{x}_n^{\mathrm{S}}, \mathbf{W})$, $n = 1, \ldots, N^{\mathrm{S}}$.

Second, the query label is predicted by Gaussian process regression with kernel function $k$ and with neural network-based mean function $g$ using the instance embeddings. The predicted label adapted to the support set is given in a closed form,

$$\hat{y}(\mathbf{x}; \mathbf{D}^{\mathrm{S}}, \mathbf{W}, \mathbf{\Theta}) = g(\mathbf{z}) + \mathbf{k}^{\top} \mathbf{K}^{-1} (\mathbf{y}^{\mathrm{S}} - \mathbf{g}^{\mathrm{S}}), \tag{4}$$

where $\mathbf{k} = [k(\mathbf{z}, \mathbf{z}_1^{\mathrm{S}}), \ldots, k(\mathbf{z}, \mathbf{z}_{N^{\mathrm{S}}}^{\mathrm{S}})] \in \mathbb{R}^{N^{\mathrm{S}}}$ is the kernel vector between the query and support instances, $\mathbf{K} \in \mathbb{R}^{N^{\mathrm{S}} \times N^{\mathrm{S}}}$ is the kernel matrix between the support instances, $\mathbf{K}_{nn'} = k(\mathbf{z}_n^{\mathrm{S}}, \mathbf{z}_{n'}^{\mathrm{S}}) \in \mathbb{R}$, $\mathbf{y}^{\mathrm{S}} = [y_1^{\mathrm{S}}, \ldots, y_{N^{\mathrm{S}}}^{\mathrm{S}}] \in \mathbb{R}^{N^{\mathrm{S}}}$ is the label vector of the support instances, and $\mathbf{g}^{\mathrm{S}} = [g(\mathbf{z}_1^{\mathrm{S}}), \ldots, g(\mathbf{z}_{N^{\mathrm{S}}}^{\mathrm{S}})] \in \mathbb{R}^{N^{\mathrm{S}}}$ is the mean vector evaluated on the support instances. $\mathbf{\Theta}$ is parameters in our model, which consists of neural network parameters in $f_{\mathrm{FE}}$, $f_{\mathrm{IE}}$, and $g$, and kernel parameters in $k$. Neural network $g$ and kernel parameters in $k$ are shared across all tasks. GP regression is a nonparametric nonlinear regression method successfully used in a wide variety of applications, and it can output prediction adapted to the support set flexibly. Since the task-adapted GP prediction is given in a closed form as shown in Eq. (4), we can efficiently perform backpropagation through the adaptation, and it has been used for meta-learning [16, 8]. The prediction in Eq. (4) can be seen as fitting GP with a zero mean function to residual $\mathbf{y}^{\mathrm{S}} - \mathbf{g}^{\mathrm{S}}$ of prediction by neural network $g$. Neural network $g$ makes predictions based on the feature descriptions without using the support set. Even when the feature descriptions are not helpful for the prediction, the proposed model makes reasonable predictions by adapting them to the support set by GP. When labels can be predicted well by neural network $g$ with the feature descriptions, since the residual becomes zero, the adaptation is skipped, by which we can avoid the risk of overfitting to the small support set.

---

**Algorithm 1** Meta-learning procedure of our model.

---

**Input:** Meta-training data $\mathcal{D}$, number of support instances $N^{\mathrm{S}}$, number of query instances $N^{\mathrm{Q}}$
**Output:** Trained model parameters $\boldsymbol{\Theta}$
 1: Initialize model parameters $\boldsymbol{\Theta}$.
 2: **while** End condition is satisfied **do**
 3:     Randomly select task index $d$ from $\{1, \cdots, D\}$.
 4:     Randomly sample $N^{\mathrm{S}}$ instances from the $d$th task for support instances $\mathbf{D}^{\mathrm{S}}$.
 5:     Randomly sample $N^{\mathrm{Q}}$ instances from the $d$th task for query instances $\mathbf{D}^{\mathrm{Q}}$ such that there is
        no overlap between the support and query instances.
 6:     Calculate test error $\frac{1}{N^{\mathrm{S}}} \sum_{(\mathbf{x},y) \in \mathbf{D}^{\mathrm{Q}}} (y - \hat{y}(\mathbf{x}; \mathbf{D}^{\mathrm{S}}, \mathbf{W}, \boldsymbol{\Theta}))^2$.
 7:     Update model parameters $\boldsymbol{\Theta}$ using the gradient of the loss by a stochastic gradient method.
 8: **end while**

---

### 3.3 Meta-training

We train model parameters $\boldsymbol{\Theta}$ by minimizing the expected test mean squared error using an episodic training framework [18],

$$\hat{\boldsymbol{\Theta}} = \arg\min_{\boldsymbol{\Theta}} \mathbb{E}_{\mathbf{D}^{\mathrm{S}}, \mathbf{D}^{\mathrm{Q}}, \mathbf{W}} \left[ \frac{1}{N^{\mathrm{Q}}} \sum_{(\mathbf{x},y) \in \mathbf{D}^{\mathrm{Q}}} (y - \hat{y}(\mathbf{x}; \mathbf{D}^{\mathrm{S}}, \mathbf{W}, \boldsymbol{\Theta}))^2 \right], \qquad (5)$$

where $\mathbb{E}_{\mathbf{D}^{\mathrm{S}}, \mathbf{D}^{\mathrm{Q}}, \mathbf{W}}$ represents the expectation over support set $\mathbf{D}^{\mathrm{S}}$, query set $\mathbf{D}^{\mathrm{Q}} = \{(\mathbf{x}_n^{\mathrm{Q}}, y_n^{\mathrm{Q}})\}_{n=1}^{N^{\mathrm{Q}}}$, and feature descriptions $\mathbf{W}$. By using the query set, which is different from the support set used for adapting GP, we can evaluate the test error. Minimizing our meta-learning loss in Eq. (5) is a bilevel optimization problem. In the inner optimization, a task-specific prediction model is adapted by minimizing the loss on the support set based on GP as described in Section 3.2.2. In the outer optimization, neural network parameters shared among all tasks are estimated by minimizing the error on the query set when the task-adapted prediction model obtained in the inner optimization is used. Algorithm 1 shows the meta-training procedure of the proposed model. The expectation in Eq. (5) is approximated by the Monte Carlo method, where a task, support and query instances are randomly sampled from the given meta-training datasets.

The computational complexity of calculating the test error for a set of support and query instances in Line 6 is $O(N^{\mathrm{Q}} + N^{\mathrm{S}^3})$ since we need the inverse of the kernel matrix whose size is $N^{\mathrm{S}} \times N^{\mathrm{S}}$. In few-shot settings, which is our focus, the number of support instances $N^{\mathrm{S}}$ is small.

### 3.4 Variants

#### 3.4.1 Feature type description

In addition to feature descriptions (e.g., "Production value at basic price" and "Wheat and spelt" in Table 1), feature type descriptions (e.g., "Agricultural indicator" and "List of products" in Table 1), are available in some datasets. Let $\mathbf{w}_j$ be the feature type description of the $j$th feature. In this case, the feature embedding is obtained by taking the sentence embeddings of the feature description and feature type description as input of the feature encoder,

$$\mathbf{u}_j = f_{\mathrm{FE}}(\mathbf{v}_{jx_j}, f_{\mathrm{SE}}(\mathbf{w}_j)), \qquad (6)$$

instead of Eq. (2). This enables us to incorporate information of feature type descriptions for meta-learning.

#### 3.4.2 Missing feature descriptions

Descriptions might be missing for a part of features in some datasets. Let $\bar{\mathbf{x}}$ be features with descriptions, and $\underline{\mathbf{x}}$ be those without descriptions in $\mathbf{x}$. Instance embedding $\mathbf{z}$ is obtained using features with descriptions by encoder $f$ in Eq. (3), $\mathbf{z} = f(\bar{\mathbf{x}})$, by which we can incorporate observed descriptions in the embedding. The task-specific GP is adapted based on the concatenation of the instance embedding and onehot vector of features without descriptions. In particular, the kernel function is calculated by, $k([\mathbf{z}, \mathrm{onehot}(\underline{\mathbf{x}})], [\mathbf{z}_{\mathrm{n}}^{\mathrm{S}}, \mathrm{onehot}(\underline{\mathbf{x}}_{\mathrm{n}}^{\mathrm{S}})])$, where onehot represents the onehot

encoder, and $[\cdot, \cdot]$ represents the concatenation of vectors. The prediction is performed by Eq. (4) using the kernel function with the concatenation, by which we can use the information on both of the features with and without descriptions for prediction, where the mean function takes instance embedding $\mathbf{z}$ as input.

### 3.4.3 Numerical feature values

When features are numerical values, we obtain the feature embedding by multiplying numerical value $x_j$ and the sentence embedding of its feature type description $\mathbf{w}_j$ as follows,

$$\mathbf{u}_j = f_{\mathrm{FE}}(f_{\mathrm{SE}}(\mathbf{w}_j)x_j). \tag{7}$$

### 3.4.4 Classification

In classification tasks, we can use the GP-based adaptation in Eq. (4) by representing labels by binary vectors as shown in [16], or we can use the Gaussian mixture model-based adaptation as in prototypical networks [20]. For the loss function in Eq. (5), we can use the cross-entropy loss instead of the mean squared error.

## 4 Experiments

### 4.1 Data

For evaluating the proposed method, we used the following two data: e-Stat and Eurostat. These data contained statistical tables in various fields, including economy, health, education, sport, and crime. The e-Stat data were obtained from the official statistics of Japan using API [1]. The Eurostat data were obtained from the statistical office of the European Union using API [2]. A statistical table contained numerical values of statistics, which were used for labels, and categorical descriptions in natural language, which were used for features, as shown in Table 1. We used a statistical table as a dataset in a task. When a statistical table comprised multiple units of measurement, it was decomposed into multiple datasets based on the unit of measurement. We used datasets where the number of instances was between 50 and 1,000. The labels were normalized with zero mean and one standard deviation for each dataset. The statistics of the two data are shown in Table 2. The descriptions were in Japanese in e-Stat, and in English in Eurostat. For e-Stat, we used the Japanese pretrained BERT [3]. For Eurostat, we used the English pretrained BERT [3, 29] [4]. The BERT vector size was 768. For each of the e-Stat and Eurostat data, we sampled 700 datasets for meta-training, 100 for meta-validation, and 200 for meta-test. For each task, the number of support instances was 5, 10, or 20, and the number of query instances was 20. We evaluated the test mean squared error averaged over ten experiments with different splits of datasets. Note that we did not use commonly-use meta-learning datasets, such as Omniglot [11], Mini-imagenet [26], and datasets in Torchmeta [2] and Meta-Dataset [23], since they do not contain feature descriptions.

### 4.2 Comparing methods

We compared the proposed method (Ours) with the following methods: ridge regression (Ridge), Gaussian process regression (GP), heterogeneous meta-learning [7] (HML), neural networks with BERT sentence embeddings (NN), meta deep kernel learning [16] with shared categorical onehot vector features (MDK+C), meta deep kernel learning with word frequency vectors (MDK+W), meta deep kernel learning with BERT sentence embeddings (MDK+B), the proposed method without mean functions (Ours-M) and the proposed method with feature type descriptions (Ours+T). Table 3 summarizes the comparing methods.

Ridge and GP methods were trained using support instances for each task. The hyperparameters, i.e., the regularization weight in Ridge and the kernel parameters in GP, were optimized using the meta-training datasets. We used RBF kernels for GP. The methods except for Ridge and GP had

---

[1] https://www.e-stat.go.jp/en.
[2] https://ec.europa.eu/eurostat/data/database.
[3] https://nlp.ist.i.kyoto-u.ac.jp/?ku_bert_japanese
[4] https://github.com/huggingface/transformers.

Table 2: Statistics of e-Stat and Eurostat data.

|  | e-Stat | Eurostat |
|---|---|---|
| Total number of tasks | 5,488 | 2,103 |
| Total number of instances | 1,379,346 | 946,959 |
| Average number of instances per task | 251.3 | 450.3 |
| Average number of features per task | 5.4 | 4.5 |
| Minimum number of features per task | 3 | 2 |
| Maximum number of features per task | 9 | 10 |
| Average number of characters per feature description | 4.9 | 16.1 |
| Total number of unique feature descriptions | 8,059 | 11,388 |
| Number of feature descriptions that appear only in one task | 3,881 | 2,278 |
| Average number of tasks per feature description | 12.6 | 18.8 |

Table 3: Comparing methods. 'Meta-learning' represents that the method has neural networks shared among tasks, which are meta-trained using the meta-training datasets. 'Task-adaptation' represents that the method adapts its prediction to the given support set by minimizing the loss. 'Feature correspondence' represents that the method uses the correspondence of features across tasks. 'Feature description' represents that the method uses feature descriptions in natural language. 'Translation' represents that the method translates the BERT sentence embedding using a feature encoder for each description. 'Pretrained BERT' represents that the method uses the pretrained BERT. 'Mean function' represents that the method uses the mean function for task-specific GP adaptation. 'Feature type description' represents that the method uses feature type descriptions.

|  | Ridge | GP | HML | NN | MDK+C | MDK+W | MDK+B | Ours-M | Ours | Ours+T |
|---|---|---|---|---|---|---|---|---|---|---|
| Meta-learning |  |  | ✓ | ✓ | ✓ | ✓ | ✓ | ✓ | ✓ | ✓ |
| Task adaptation | ✓ | ✓ |  |  | ✓ | ✓ | ✓ | ✓ | ✓ | ✓ |
| Feature correspondence |  |  | ✓ | ✓ | ✓ | ✓ | ✓ | ✓ | ✓ | ✓ |
| Feature description |  |  | ✓ | ✓ |  | ✓ | ✓ | ✓ | ✓ | ✓ |
| Pretrained BERT |  |  | ✓ |  |  |  | ✓ | ✓ | ✓ | ✓ |
| Translation |  |  | ✓ |  |  | ✓ |  | ✓ | ✓ | ✓ |
| Mean function |  |  | ✓ | ✓ | ✓ | ✓ | ✓ |  | ✓ | ✓ |
| Feature type description |  |  |  |  |  |  |  |  |  | ✓ |

neural networks shared among all tasks, and they were meta-trained using the meta-training datasets. HML was a meta-learning method for tasks with different feature spaces, which did not use feature information. Note that since other existing meta-learning methods cannot handle tasks with different feature spaces, we did not compare with them. In Ridge, GP, and HML, categorical feature values were transformed into onehot vectors for each task. NN was a neural network-based meta-learning method, where feature descriptions were used for its input. With NN, feature descriptions were embedded by BERT, translated by a feature encoder network, and taken to deep sets [32] as input for predicting labels. NN corresponds to the proposed method without GP adaptation to the support set for each task. In MDK+C, MDK+W and MDK+B, a task-specific GP was adapted using the support set, where RBF kernels were used with neural networks for input transformation. MDK+C used onehot vectors of categorical feature values that were shared in all tasks, MDK+C corresponds to the proposed method that uses feature descriptions for only identifying feature correspondences across different tasks. MDK+W used word frequency vectors for representing feature descriptions, where the vectors were embedded in a 768-dimensional space by principal component analysis. MDK+W corresponds to the proposed method with word frequency vectors without pretrained BERT. MDK+B used BERT sentence embeddings, but they were not translated by a feature encoder for each description in meta-learning. MDK+B corresponds to the proposed method without feature encoders. Since the feature type descriptions were unavailable by Eurostat API, we did not evaluate Ours+T on Eurostat data.

## 4.3 Settings

In the proposed model, we used a three-layered feed-forward neural network with 128 hidden and output units for the sentence, feature, and instance encoders, $f_{\text{SE}}$, $f_{\text{FE}}$, and $f_{\text{IE}}$, and a three-layered feed-forward neural network with 128 hidden units and a single output unit for mean function $g$. For the activation function, we used rectified linear unit $\text{ReLU}(x) = \max(0, x)$. For GP, we used RBF

Table 4: Average test mean squared error and its standard error with different numbers of support instances per task. Values in bold typeface are not statistically significantly different at the 5% level from the best performing method in each row according to a paired t-test.

(a) e-Stat

| #Support | 5 | 10 | 20 |
|---|---|---|---|
| Ridge | $0.991 \pm 0.015$ | $0.889 \pm 0.019$ | $0.663 \pm 0.017$ |
| GP | $0.921 \pm 0.016$ | $0.807 \pm 0.018$ | $0.626 \pm 0.016$ |
| HML | $0.860 \pm 0.017$ | $0.857 \pm 0.039$ | $0.652 \pm 0.077$ |
| NN | $0.462 \pm 0.018$ | $0.462 \pm 0.018$ | $0.462 \pm 0.018$ |
| MDK+C | $0.607 \pm 0.017$ | $0.583 \pm 0.013$ | $0.555 \pm 0.008$ |
| MDK+W | $0.470 \pm 0.016$ | $0.413 \pm 0.022$ | $\mathbf{0.331 \pm 0.009}$ |
| MDK+B | $0.444 \pm 0.013$ | $0.406 \pm 0.013$ | $0.350 \pm 0.007$ |
| Ours-M | $0.658 \pm 0.025$ | $0.490 \pm 0.020$ | $0.366 \pm 0.011$ |
| Ours | $0.448 \pm 0.014$ | $\mathbf{0.383 \pm 0.013}$ | $\mathbf{0.320 \pm 0.010}$ |
| Ours+T | $\mathbf{0.417 \pm 0.011}$ | $\mathbf{0.375 \pm 0.012}$ | $\mathbf{0.333 \pm 0.015}$ |

(b) Eurostat

| #Support | 5 | 10 | 20 |
|---|---|---|---|
| Ridge | $0.969 \pm 0.021$ | $0.925 \pm 0.015$ | $0.790 \pm 0.017$ |
| GP | $0.902 \pm 0.018$ | $0.844 \pm 0.016$ | $0.749 \pm 0.018$ |
| HML | $0.961 \pm 0.018$ | $0.934 \pm 0.029$ | $0.963 \pm 0.025$ |
| NN | $0.896 \pm 0.014$ | $0.896 \pm 0.014$ | $0.896 \pm 0.014$ |
| MDK+C | $0.885 \pm 0.022$ | $0.851 \pm 0.019$ | $0.731 \pm 0.013$ |
| MDK+W | $0.920 \pm 0.017$ | $0.843 \pm 0.022$ | $0.701 \pm 0.015$ |
| MDK+B | $0.858 \pm 0.016$ | $0.762 \pm 0.015$ | $0.671 \pm 0.017$ |
| Ours-M | $0.867 \pm 0.029$ | $0.781 \pm 0.019$ | $0.634 \pm 0.023$ |
| Ours | $\mathbf{0.756 \pm 0.021}$ | $\mathbf{0.690 \pm 0.019}$ | $\mathbf{0.595 \pm 0.019}$ |

kernels, $k(\mathbf{z}, \mathbf{z}') = \alpha \exp\left(-\frac{\gamma}{2} \parallel \mathbf{z} - \mathbf{z}' \parallel^2\right) + \beta\delta(\mathbf{z}, \mathbf{z}')$, where $\alpha$, $\beta$, and $\gamma$ were kernel parameters to be meta-trained. We optimized our models using Adam [10] with learning rate $10^{-3}$, batch dataset size 32, and dropout rate 0.1 [22]. The meta-validation datasets were used for early stopping, for which the maximum number of meta-training epochs was 5,000. In the comparing methods, we used the same neural network architectures and training procedures as the proposed method. We implemented the proposed method with PyTorch [15].

## 4.4 Results

Table 4 shows the test mean squared error with different numbers of support instances per task. The proposed method achieved the best performance in all cases. The performance of the Ridge and GP methods was low since they were trained using only support instances in the meta-test datasets. The HML method did not perform well since it needs to meta-learn relationships among tasks using support instances without feature descriptions. This result indicates the effectiveness of the feature descriptions for meta-learning. The NN method predicts labels using only feature descriptions, where the prediction is not adapted to the labeled support instances. Since the feature descriptions in natural language are often not quantitative, it is difficult for the NN method to predict labels accurately. On the other hand, the proposed method achieved better performance than the NN method by minimizing the prediction error on the support instances. The MDK+C method uses feature descriptions for corresponding features among tasks, but does not use sentence embeddings. The MDK+C method performed worse than the proposed method, which implies sentence embeddings are important for meta-learning since they can quantify the relationships among features even when their descriptions are not the same. The MDK+W method underperformed the proposed method, which implies that incorporating knowledge in large-scale corpus stored by pretrained BERT is effective for meta-learning. The error by the MDK+B method was higher than the proposed method, which indicates the effectiveness of the translation using the feature encoder for meta-learning. The proposed method without mean functions (Ours-M) performed worse than the NN method on e-Stat data with five and ten support instances. It is because feature descriptions are relatively important for task-specific prediction when the number of support instances is small. With the proposed method

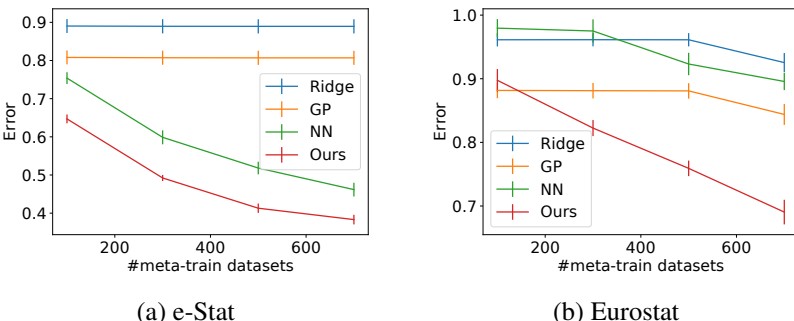

(a) e-Stat  (b) Eurostat

Figure 3: Average test mean squared errors with different numbers of meta-training datasets. The bar shows the standard error.

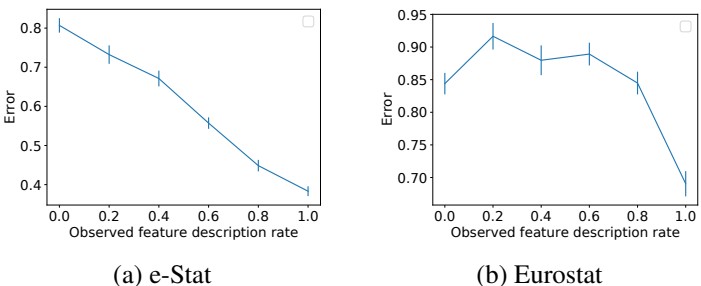

(a) e-Stat  (b) Eurostat

Figure 4: Average test mean squared errors with different rates of observed feature descriptions. The bar shows the standard error.

Table 5: Average test mean squared error and its standard error by the proposed method with word2vec instead of BERT.

| #Support | 5 | 10 | 20 |
|---|---|---|---|
| e-Stat | $0.500 \pm 0.016$ | $0.442 \pm 0.012$ | $0.390 \pm 0.013$ |
| Eurostat | $0.865 \pm 0.021$ | $0.800 \pm 0.016$ | $0.680 \pm 0.017$ |

(Ours), the residual error of the neural network-based mean function is adapted by Gaussian processes using support instances for each task, which enables us to achieve better performance for all the cases. With the proposed method, the use of feature type descriptions (Ours+T) improved the performance when the number of support instances was five. The feature type descriptions can be helpful when support instances are small.

Figure 3 shows the test mean squared error with different numbers of meta-training datasets. As the number of the meta-training datasets increased, the performance of the proposed method increased. The proposed method achieved the best performance in all cases except for 100 meta-training datasets in Eurostat data. It is effective for the proposed method to use more meta-training datasets. Figure 4 shows the test mean squared error with different rates of observed feature descriptions by the proposed method. The error decreased as the observed rate increased especially in e-Stat data.

Table 5 shows the test mean squared error by the proposed method using word2vec [12] for sentence embeddings instead of BERT. We used Japanese word2vec [5] for e-Stat, and English word2vec [6] for Eurostat, and the sentence embedding was obtained by the average of the word embeddings in the description. The error with word2vec was higher than that with BERT (Ours in Table 4). This result indicates that the meta-learning performance depends on the performance of the sentence encoder.

Table 6 shows the test mean squared error on tasks where more than half of the feature descriptions do not appear in the meta-training tasks. The proposed method also achieved lower errors in tasks that contain feature descriptions that do not appear in the meta-training data than the other methods.

---

[5] https://github.com/Kyubyong/wordvectors
[6] https://code.google.com/archive/p/word2vec

Table 6: Average test mean squared errors on tasks where more than half of the feature descriptions do not appear in the meta-training tasks with five support instances per task.

|  | Ridge | GP | HML | NN | MDK+C | MDK+W | MDK+B | Ours-M | Ours | Ours+T |
|---|---|---|---|---|---|---|---|---|---|---|
| e-Stat | 0.968 | 0.892 | 0.836 | 0.613 | 0.665 | 0.747 | 0.685 | 0.830 | **0.607** | 0.568 |
| Eurostat | 1.019 | 0.942 | 0.993 | 1.013 | 0.987 | 0.993 | 1.023 | 0.951 | **0.866** | |

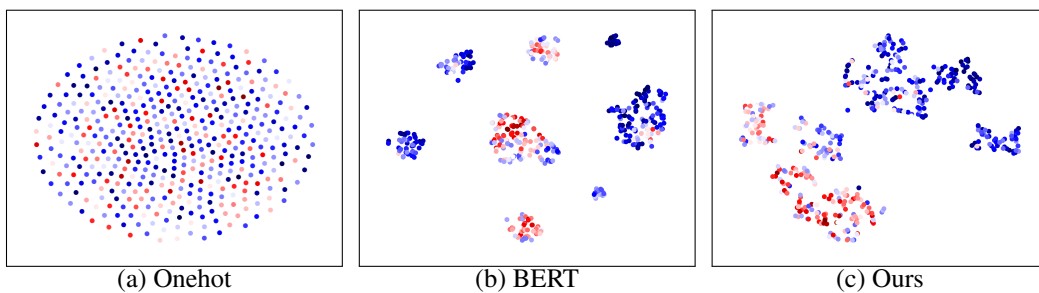

| (a) Onehot | (b) BERT | (c) Ours |
|---|---|---|

Figure 5: Visualization by t-SNE [24] of instances in a task on e-Stat data. Each point is an instance, and its color represents the label. Each instance is represented by (a) a onehot vector of categorical feature values, (b) a concatenation of BERT sentence embeddings of feature descriptions, and (c) an instance embedding with the proposed method.

Table 7: Meta-training time in seconds on e-Stat data with ten support instances per task.

| Ridge | GP | HML | NN | MDK+C | MDK+W | MDK+B | Ours-M | Ours | Ours+T |
|---|---|---|---|---|---|---|---|---|---|
| 3113.1 | 2695.5 | 92664.6 | 13642.3 | 21134.2 | 14469.3 | 11775.5 | 14661.8 | 16841.4 | 24483.3 |

Figure 5 shows the visualization of instances. With the onehot vectors of categorical feature values (a), it is difficult to represent the instances well. With the BERT sentence embeddings (b), instances with similar labels formed clusters, but the clusters were separated. With the proposed method (c), clusters with similar labels were located more closely than those with the BERT sentence embeddings by the feature and instance encoders, which helps label prediction in the instance embedding space.

With translation by the feature encoder, similar BERT sentence embeddings can be encoded to different feature embeddings for improving the meta-learning performance. For example, in Eurostat data, BERT embeddings of 'Accommodation (10 or more employees and self-employed persons)' and 'Information and communication (10 or more employees and self-employed persons)' were similar (cosine=0.943) since the words in the parenthesis were the same, while their translated feature embeddings by the proposed method were different (cosine=0.003). BERT embeddings of 'Outwards' and 'Inwards' were similar (cosin=0.951), while their feature embeddings were different (cosine=0.025), since they were related but antonyms. BERT embeddings of 'Warmond - Hoofddorp' and 'Amersfoort - Hilversum' were different (cosine=0.280), while their feature embeddings were similar (cosine=0.901) since they are towns in the Netherland and related.

Table 7 shows the average computation time in seconds for meta-training on computers with 2.60GHz CPUs. The Ridge and GP methods were efficient since they did not have neural networks to be meta-trained. The computational time of the proposed method was shorter than that of the HML method.

## 5   Conclusion

We proposed a meta-learning method that effectively uses feature descriptions, and confirmed that the proposed method achieves better performance than the existing methods. Although our results are encouraging as an effective approach for sharing knowledge between various tasks in meta-learning by incorporating human knowledge, there are some limitations to overcome. First, we need to empirically evaluate the variants of the proposed method, e.g., for classification tasks and numerical feature values. Second, we would like to use other types of neural networks for components in our model, such as attention networks [25], since we used feed-forward neural networks in our experiments.

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
