# Supplemental material: Sharing Knowledge for Meta-learning with Feature Descriptions

## 1 Experimental results

Figure 1 shows the test mean squared error with different rates of observed feature descriptions by the proposed method. The error decreased as the observed rate increased especially in e-Stat data.

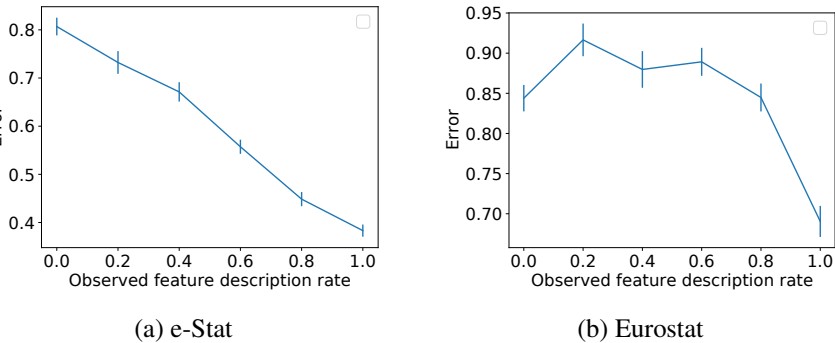

| (a) e-Stat | (b) Eurostat |
|------------|--------------|

Figure 1: Average test mean squared errors with different rates of observed feature descriptions. The bar shows the standard error.

Table 1 shows the test mean squared error by the proposed method using word2vec [1] for sentence embeddings instead of BERT. We used `https://github.com/Kyubyong/wordvectors` for e-Stat, and `https://code.google.com/archive/p/word2vec` for Eurostat, and the sentence embedding was obtained by the average of the word embeddings in the sentence. The error with word2vec was higher than that with BERT. This result indicates that the meta-learning performance depends on the performance of the sentence encoder.

Table 1: Average test mean squared error and its standard error by the proposed method with word2vec instead of BERT.

|  (a) e-Stat | | | |
|-------------|-----|-----|-----|
| #Support | 5 | 10 | 20 |
| Ours+word2vec | $0.500 \pm 0.016$ | $0.442 \pm 0.012$ | $0.390 \pm 0.013$ |
|  (b) Eurostat | | | |
| #Support | 5 | 10 | 20 |
| Ours+word2vec | $0.865 \pm 0.021$ | $0.800 \pm 0.016$ | $0.680 \pm 0.017$ |

## 2  Limitations

The proposed method requires feature descriptions for meta-training datasets. When there are not enough meta-training datasets, the performance can be comparable with methods that are trained using support instances for each task. When meta-training and meta-test tasks are significantly different, there is a risk of degrading the performance. The evaluation of the proposed method with classification tasks and numerical feature values are not performed since the data used in our experiments were regression tasks with categorical feature values.

## 3  Potential negative societal impacts

Since the proposed method uses various datasets for meta-training, there is a potential risk that users might include biased datasets without careful thought, which might result in biased predictions. We encourage research to automatically detect biased datasets. Although the proposed method improves performance, the prediction might be not always correct. We encourage research to estimate the uncertainty of the prediction by meta-learning from datasets with feature descriptions.

## References

[1] T. Mikolov, K. Chen, G. Corrado, and J. Dean. Efficient estimation of word representations in vector space. *arXiv preprint arXiv:1301.3781*, 2013.