# OpenReview forum: "Sharing Knowledge for Meta-learning with Feature Descriptions"
_NeurIPS.cc/2022/Conference — NeurIPS 2022 Accept_

### Official Review · Reviewer_Ay7F · 2022-07-11

**Rating:** 7
**Confidence:** 3
**Soundness:** 4 excellent
**Presentation:** 3 good
**Contribution:** 4 excellent

**Summary:**

This paper considers the problem of heterogenous meta learning, where the different datasets potentially have different number and type of features.
To enable using the same model for all the datasets, the authors consider an approach where they use textual feature descriptions, which makes the model permutation invariant with respect to the features and also agnostic to the number of features.
This allows the same model to be used for all tasks, regardless of their composition of features, because new features can simply be encoded using a sentence encoder.
They show that they outperform other approaches on two datasets.

**Questions:**

1. Is `Ours + T` the same as `Ours + F` in Table 4?
1. How do you think the model would fare if you directly used the instance embeddings (`z`) to make the prediction through a function approximator which is fine-tuned on the support set rather using a GP. In other words, what if you did few-shot fine-tuning on the support set? I believe this would be equivalent to `NN` with few-shot fine-tuning.

**Limitations:**

Has been addressed in the supplementary PDF.

**Strengths And Weaknesses:**

**Strengths**
1. The approach is very sound and in line with other recent approaches which propose to use descriptions of classes, features, and tasks.
1. The method was compared with a lot of other baselines which helps put the scores in context.
1. The gains seems to be consistent and strong on both meta-tasks considered.


**Weaknesses**
1. It looks like the only difference between the proposed method and a baseline (`MDK + B`) is the usage of the feature encoder (`Fig 1`), which is a 3 layer neural network. It looks like the authors agree with that as well (line 220). So the technical novelty (although guided by good intuition) seems to boil down to the addition of a single layer on top of a baseline.
1. I think it is important to experiment with different types of sentence encoders. Given that the descriptions they consider are very short, it's possible that simple word2vec vectors (bag-of-words averaging) could do the trick.
1. Another baseline could have been added, which fine-tunes the `NN` model on the downstream task.

---

> ### Author Response · Authors · 2022-08-02
> **Response**
>
> Thank you for your constructive comments.
>
> > It looks like the only difference between the proposed method and a baseline (MDK + B) is the usage of the feature encoder (Fig 1), which is a 3 layer neural network. It looks like the authors agree with that as well (line 220). So the technical novelty (although guided by good intuition) seems to boil down to the addition of a single layer on top of a baseline.
>
> MDK+B is not an existing method. We compared with MDK+B to demonstrate the effectiveness of the feature encoder in the proposed method. The original meta deep kernel learning (MDK) [13] cannot use feature descriptions, cannot handle tasks with different number of features, and cannot be used for our problem. Therefore, the technical novelty is not just the addition of a single layer on top of an existing method.
>
> > I think it is important to experiment with different types of sentence encoders. Given that the descriptions they consider are very short, it's possible that simple word2vec vectors (bag-of-words averaging) could do the trick.
>
> MDK+W can be think as the proposed method with a different type of (very simple) sentence encoders, where principal component analysis is used for the sentence encoder. We did not conduct experiments with other types of sentence encoders since any sentence encoders can be used in our framework. We will conduct experiments with different sentence encoders (e.g., word2vec) to check the sensitivity of sentence encoders.
>
> > Another baseline could have been added, which fine-tunes the NN model on the downstream task.
>
> We did not include fine-tunes the NN model on the downstream task for comparing methods, because such an approach does not work well when the number of labeled instances is very small, which is our focus, as shown in existing meta-learning papers (e.g., [3]).
>
> > Is Ours + T the same as Ours + F in Table 4?
>
> Yes. We will fix it.
>
> > How do you think the model would fare if you directly used the instance embeddings (z) to make the prediction through a function approximator which is fine-tuned on the support set rather using a GP. In other words, what if you did few-shot fine-tuning on the support set? I believe this would be equivalent to NN with few-shot fine-tuning.
>
> We think that such an approach can be used for our problem (we understand that it uses model-agnostic meta-learning (MAML) [3] instead of meta deep kernel learning (MDK) [13] in our framework). Our model can be think that the last layer of NN based on GP is fine-tuned using the support set. We used GPs since MDK works better than MAML in [13].

---

### Official Review · Reviewer_CRpu · 2022-07-12

**Rating:** 4
**Confidence:** 4
**Soundness:** 3 good
**Presentation:** 3 good
**Contribution:** 2 fair

**Summary:**

This work proposes to use the BERT representation of the textual description of the feature value as input feature to improve the model generalization, particularly in the meta-learning setup. Empirically, compared with baseline meta-learning methods on 2 datasets, the proposed method achieves better or competitive performances.

**Questions:**

- For categorical features that do not come with descriptions, is there any systematic way to generate the feature description?
- Since the feature description is manually written, will the model quality be sensitive to the choice of the description?

**Limitations:**

The work only tests the proposed method in two relatively small datasets. Hence, it's hard to judge how general the conclusions are.

**Strengths And Weaknesses:**

The idea of the work is very interesting and intuitive. The paper is well written and easy to understand. The empirical evaluation shows some promising results.

However, the empirical evaluation is done only on two relatively small datasets, which makes it difficult to judge how general the proposed method is in practice. It would be much more informative to try this method on large datasets.

---

> ### Author Response · Authors · 2022-08-02
> **Response**
>
>
> Thank you for your constructive comments.
>
> > the empirical evaluation is done only on two relatively small datasets, which makes it difficult to judge how general the proposed method is in practice. It would be much more informative to try this method on large datasets.
>
> We chose these two datasets since they contain feature descriptions, and many tasks that are necessary for meta-learning. The total number of instances in the meta-training data is 175,910 in e-Stat and 315,210. They are not small compared with commonly used meta-learning datasets (e.g., miniImageNet contains 50,000 training images, and Omniglot contains 32,460 handwritten characters). We did not use commonly used datasets in the existing meta-learning literature, e.g., Omniglot, Mini-imagenet, and datasets in Meta-Dataset[20], because they do not contain feature descriptions as described at the last sentence of Sec 4.1.
>
> We believe that the better performance of the proposed method compared with the comparing method in Table 4 demonstrates the effectiveness of each component in the proposed method. The decrease of the error as the number of meta-training datasets in Figure 3 demonstrates that the meta-learning of the proposed method works well.
>
> > For categorical features that do not come with descriptions, is there any systematic way to generate the feature description?
>
> The feature descriptions might be generated by supervised learning to predict the description given categorical features, or other features. Although we do not know existing work for generating the feature descriptions, if it exists, we can use it before applying the proposed method.
>
> > Since the feature description is manually written, will the model quality be sensitive to the choice of the description?
>
> If the pretrained BERT gives appropriate representations for descriptions (e.g., even if descriptions are not the same but have similar meaning, their representations are similar), or if the sentence encoder is trained well by meta-training data, the choice of the description would not be so sensitive.

---

### Official Review · Reviewer_UMH4 · 2022-07-12

**Rating:** 6
**Confidence:** 3
**Soundness:** 3 good
**Presentation:** 3 good
**Contribution:** 3 good

**Summary:**

The paper proposes a meta-learning method that builds relationships across supervised learning tasks with different feature spaces using feature descriptions written in natural language. When different tasks may share similar or related feature descriptions, benefiting from the pre-trained language model (e.g., BERT), which contains different kinds of knowledge implicitly, the proposed method could improve the generalization performance on unseen tasks with small, labeled data. The authors also empirically demonstrate that the proposed method outperforms the existing meta-learning methods in real-world datasets.

**Questions:**

Whether the feature description sets or feature type sets exist intersection between meta-training datasets and meta-test datasets？

**Ethics Review Area:**

["I don’t know"]

**Limitations:**

The  authors have addressed the limitations.

**Strengths And Weaknesses:**

Strengths:
1. The novel problem setting is reasonable and challenging, where instances have the same feature space in the same task and different feature description sets and feature spaces among different tasks.

2. Extensive experiments and ablation studies are provided to determine the effectiveness of the proposed method.

3. The paper is well-written and easy to follow.

Weaknesses:
1. Despite the plausible setting, there are still a few points that the authors fail to explain elaborately. For example, whether the feature description sets or feature type sets exist intersection between meta-training datasets and meta-test datasets. The question is out of my curiosity about whether the meta-training model using feature descriptions could really generalize to those unseen tasks which contain entirely different feature description sets or even feature types from the meta-training tasks.

2. As the authors mentioned in the Limitation section, the proposed method exits several limitations. First, the method seems to depend on the number of meta-training datasets. Second, in this work, it is to be doubted whether the method could extend to those datasets, where instances contain numerical features or the corresponding labels are discrete. It could be better to find a dataset conforming to the characteristics described above to evaluate the effectiveness of the method.

3. A baseline that neural network with feature description is not meta-trained could be considered, which could evaluate the necessity of meta-training despite using the feature description.

---

> ### Author Response · Authors · 2022-08-02
> **Response**
>
> Thank you for your constructive comments.
>
> > Despite the plausible setting, there are still a few points that the authors fail to explain elaborately. For example, whether the feature description sets or feature type sets exist intersection between meta-training datasets and meta-test datasets. The question is out of my curiosity about whether the meta-training model using feature descriptions could really generalize to those unseen tasks which contain entirely different feature description sets or even feature types from the meta-training tasks.
>
> There are intersections of feature descriptions between meta-training and meta-test datasets. For example, country names and years appear in most tasks in Eurostat. In Eurostat, there were 8059 unique feature descriptions. 3881 out of the 8059 appeared only in one task. The average number of tasks that a unique feature description appears is 13 tasks. We will add explanations on the datasets.
>
> Since some of the feature descriptions (e.g., country names and years) appear in many tasks, we cannot show the performance on meta-test tasks with entirely different feature description sets. Instead, the following table shows the test error on tasks where more than half of the feature descriptions do not appear in the meta-training tasks. The proposed method also achieved lower errors in tasks that contain feature descriptions that do not appear in the meta-training data than the other methods. We will add these analysis.
>
> Table: Average test mean squared errors on tasks where more than half of the feature descriptions do no appear in the meta-training tasks (#support=5).
> ||Ridge|GP|HML|NN|MDK+C|MDK+W|MDK+B|Ours-M|Ours|Ours+F|
> |:----|----:|----:|----:|----:|----:|----:|----:|----:|----:|---:|
> |e-Stat  |0.968|0.892|0.836|0.613|0.665|0.747|0.685|0.830|0.607|0.568|
> |Eurostat|1.019|0.942|0.993|1.013|0.987|0.993|1.023|0.951|0.866|     |
>
> > First, the method seems to depend on the number of meta-training datasets. Second, in this work, it is to be doubted whether the method could extend to those datasets, where instances contain numerical features or the corresponding labels are discrete. It could be better to find a dataset conforming to the characteristics described above to evaluate the effectiveness of the method.
>
> We admit that the method depends on the number of meta-training datasets as described in the limitation section. Many existing meta-learning methods also have this limitation. Although we described how to handle numerical features and discrete labels in Section 3.4, we have not evaluated how they work in our experiments since we cannot find datasets with many tasks of numerical features or discrete labels with feature descriptions. Despite these limitations, we believe our contributions are important for the study to share knowledge in various tasks in meta-learning.
>
> > A baseline that neural network with feature description is not meta-trained could be considered, which could evaluate the necessity of meta-training despite using the feature description.
>
> NN in our experiments corresponds to the neural network with feature description without meta-training. The better performance of the proposed method compared with NN demonstrates the effectiveness of meta-learning in the proposed method.
>
> > Whether the feature description sets or feature type sets exist intersection between meta-training datasets and meta-test datasets.
>
> As described above, there are intersections of feature descriptions and types between meta-training datasets and meta-test datasets.

---

### Official Review · Reviewer_D33L · 2022-07-12

**Rating:** 5
**Confidence:** 3
**Soundness:** 3 good
**Presentation:** 3 good
**Contribution:** 2 fair

**Summary:**

This paper proposes a meta-learning method that uses natural language to describe features. The natural language feature description can be encoded with pretrained models in the same way for different tasks, thus enabling the transferability of these features across tasks. The author conducted experiments using some statistical datasets that have categorical features and numerical labels and split these datasets into subsets for meta training, meta validation, and meta testing. Experiments show that the proposed method achieve faster adaptation to a new task with a few task-specific examples.

**Questions:**

See the weaknesses above.

**Limitations:**

Yes, the author discussed the limitations quite well.

**Strengths And Weaknesses:**

Strengths:
1. The idea of describing features in natural language in the meta-learning setting looks novel and reasonable because such natural language features can be easily sharable.
2. The problem formulation and method description are very clear.

Weaknesses:
1. Despite the idea being general, the evaluation setting looks very restricted. The datasets are not commonly used in the existing meta-learning literature as far as I know. I would suggest the author justify why they chose these datasets, and test their method more broadly on other datasets too.
2. It's not clear what the tasks are in their datasets and how related they are. How different are the features of different tasks? The author should give some examples for readers to better understand the difficulty of the generalization across these tasks.
3. From the results in Table 4, it seems most of the improvement is from using pretrained models (comparing NN v.s. ours), which makes me question the contributions of other parts of the proposed method.
4. Regarding technical details, the author averages the embeddings of different features (equation 3), which raises a concern about how representative the resulted vector can be if the number of features increases. In this work, it seems not a big issue because the number of features for each task is very small (4-5), but I question this if the proposed method is used for many real machine learning problems which usually have thousands of features.

---

> ### Author Response · Authors · 2022-08-02
> **Response**
>
> Thank you for your constructive comments.
>
> > Despite the idea being general, the evaluation setting looks very restricted. The datasets are not commonly used in the existing meta-learning literature as far as I know. I would suggest the author justify why they chose these datasets, and test their method more broadly on other datasets too.
>
> We did not use commonly used datasets in the existing meta-learning literature, e.g., Omniglot, Mini-imagenet, and datasets in Meta-Dataset[20], because they do not contain feature descriptions as described at the last sentence of Sec 4.1. We used e-Stat and Eurostat since they contain feature descriptions, and there are many tasks that are necessary for meta-learning. We will add more justification why we chose these datasets in the revised paper.
>
> > It's not clear what the tasks are in their datasets and how related they are. How different are the features of different tasks? The author should give some examples for readers to better understand the difficulty of the generalization across these tasks.
>
> In Eurostat, there were 8059 unique feature descriptions. 3881 out of the 8059 appeared only in one task. The average number of tasks that a unique feature description appears is 13 tasks. The difficulty of the generalization across these tasks is shown by the low performance of the comparing methods. For example, due to the difficulty of the generalization across tasks without feature descriptions, the error by HML was high. The low performance of NN (using BERT representations of feature descriptions as input of a neural network) also shows the difficulty. We will add examples to show how the features of different tasks are different.
>
> > From the results in Table 4, it seems most of the improvement is from using pretrained models (comparing NN v.s. ours), which makes me question the contributions of other parts of the proposed method.
>
> The better performance of Ours compared with NN demonstrates the effectiveness of the task adaptation described in Sec 3.2.2. Table 3 explains the difference of Ours and the other comparing methods. The better performance of Ours compared with MDK+W shows the effectiveness of the use of pretrained BERT. The better performance compared with MDK+B shows the contribution of the translation by the sentence encoder in Eq.(1). The better performance compared with Ours-M shows the contribution of the mean function in Eq.(4). We will clarify the contribution of each part of the proposed method.
>
> > Regarding technical details, the author averages the embeddings of different features (equation 3), which raises a concern about how representative the resulted vector can be if the number of features increases. In this work, it seems not a big issue because the number of features for each task is very small (4-5), but I question this if the proposed method is used for many real machine learning problems which usually have thousands of features.
>
> As shown in Deep sets' paper [29], permutation invariant functions can be represented by $\rho(\sum_x \phi(x))$ using suitable transformation $\phi$ and $\rho$. We use neural networks $f_{FE}$ and $f_{IE}$ before and after the summation, which is a similar structure with the deep set. We used the average instead of the summation because the average is often more stable than the summation. The averaging has been successfully used for modeling permutation invariant functions, which include cases with a large number of embeddings (e.g., [Garnelo et al, Conditional Neural Processes, ICML, 2018]). The representation power can be improved by using attention networks for $f_{FE}$ before the averaging, which is one of our future works as described in Conclusion.

---

### Meta-Review · Area_Chair_xY3Q · 2022-08-31

**Recommendation:** Accept
**Confidence:** Less certain

**Metareview:**

This paper presents a novel meta-learning approach based on learning a sentence encoder which maps feature descriptions to embeddings. The sentence encoder is shown to generalize to new tasks during the test phase, hence allowing few-shot learning. The main concern raised by the reviewers was about the use of only two datasets which are non-standard for evaluation meta-learning. However, as the authors note, the proposed approach requires using datasets where feature descriptions are available and hence the choice of datasets seems reasonable. The authors are encouraged to revise the paper to discuss how the approach might be generalized other setups in meta-learning.

**Award:**

No

---

### Decision · Program_Chairs · 2022-09-14

Accept